# Therapeutic Targeting of Alternative RNA Splicing in Gastrointestinal Malignancies and Other Cancers

**DOI:** 10.3390/ijms222111790

**Published:** 2021-10-30

**Authors:** Ilyas Sahin, Andrew George, Attila A. Seyhan

**Affiliations:** 1Division of Hematology Oncology, Department of Medicine, University of Florida Health Cancer Center, Gainesville, FL 32610, USA; i.sahin@ufl.edu; 2Department of Chemistry, Brown University, Providence, RI 02912, USA; andrew_george@brown.edu; 3Department of Molecular Biology, Cell Biology & Biochemistry, Division of Biology and Medicine, Brown University, Providence, RI 02912, USA; 4Laboratory of Translational Oncology and Experimental Cancer Therapeutics, Warren Alpert Medical School, Brown University, Providence, RI 02912, USA; 5The Joint Program in Cancer Biology, Brown University and Lifespan Health System, Providence, RI 02912, USA; 6Cancer Center at Brown University, Warren Alpert Medical School, Brown University, Providence, RI 02912, USA; 7Department of Pathology and Laboratory Medicine, Warren Alpert Medical School, Brown University, Providence, RI 02912, USA

**Keywords:** dysregulation of RNA processing, alternative splicing, therapeutic targeting of alternative splicing, cancer, gastrointestinal malignancies

## Abstract

Recent comprehensive genomic studies including single-cell RNA sequencing and characterization have revealed multiple processes by which protein-coding and noncoding RNA processing are dysregulated in many cancers. More specifically, the abnormal regulation of mRNA and precursor mRNA (pre-mRNA) processing, which includes the removal of introns by splicing, is frequently altered in tumors, producing multiple different isoforms and diversifying protein expression. These alterations in RNA processing result in numerous cancer-specific mRNAs and pathogenically spliced events that generate altered levels of normal proteins or proteins with new functions, leading to the activation of oncogenes or the inactivation of tumor suppressor genes. Abnormally spliced pre-mRNAs are also associated with resistance to cancer treatment, and certain cancers are highly sensitive to the pharmacological inhibition of splicing. The discovery of these alterations in RNA processing has not only provided new insights into cancer pathogenesis but identified novel therapeutic vulnerabilities and therapeutic opportunities in targeting these aberrations in various ways (e.g., small molecules, splice-switching oligonucleotides (SSOs), and protein therapies) to modulate alternative RNA splicing or other RNA processing and modification mechanisms. Some of these strategies are currently progressing toward clinical development or are already in clinical trials. Additionally, tumor-specific neoantigens produced from these pathogenically spliced events and other abnormal RNA processes provide a potentially extensive source of tumor-specific therapeutic antigens (TAs) for targeted cancer immunotherapy. Moreover, a better understanding of the molecular mechanisms associated with aberrant RNA processes and the biological impact they play might provide insights into cancer initiation, progression, and metastasis. Our goal is to highlight key alternative RNA splicing and processing mechanisms and their roles in cancer pathophysiology as well as emerging therapeutic alternative splicing targets in cancer, particularly in gastrointestinal (GI) malignancies.

## 1. Introduction

Cancer is a complex and heterogeneous disease that evolves through successive genetic and epigenetic changes that support tumorigenesis [1]. These genetic and epigenetic changes often result in the activation of oncogenes and the suppression of tumor suppressor genes constitutively in conditions in which their wild-type counterparts are not, and inactivate tumor-suppressor genes [1].

Changes in the genome that affect gene function often result from various genetic and genomic abnormalities including chromosomal translocations, insertions or deletions, amplifications, and single-nucleotide mutations or alterations in the epigenome as well as the dysregulation of specific suppressor miRs or oncomiRs; the upregulation or downregulation of global miRNA levels as a consequence of dysregulated miRNA biogenesis pathways also play a role in cancer pathogenesis [2,3]. In addition, pre-mRNAs generated from the transcription of protein-coding genes are subjected to a series of chemical and structural modifications, such as the removal of introns by splicing, cleavage of mRNA at the 3′ end, the addition of a long chain of adenine nucleotides known as the poly(A) tail to form mature mRNA in the nucleus, the subsequent exportation to the cytoplasm, and the translation into the protein that they code for [1].

More recently, large-scale comprehensive genomic studies including single-cell RNA sequencing and characterization have revealed multiple processes by which protein-coding and noncoding RNA processing are dysregulated in many cancers. Among these, mutations that drive cancer by perturbing co-transcriptional and post-transcriptional regulation of gene expressions, such as alterations that affect each phase of RNA processing, including the transcription, splicing, transport, editing, and decay of protein-coding and noncoding RNAs, including microRNAs (miRNAs), have been implicated in the pathogenesis of many cancers.

As a disease with incredible complexity in its biochemical and genetic landscape, it is unsurprising that recent scientific progress has highlighted the importance of understanding the aberrant nature of mRNA processing, more specifically, alternate RNA splicing, as an intermediary in gene expression in various types of cancers. Alternate RNA splicing was initially discovered in 1977 as a mechanism of protein diversity, allowing multiple variants of a single mRNA molecule to be produced by processing in eukaryotic cells [4,5], and later work has confirmed the significance of this mechanism in protein production, playing a role in approximately 95% of multiexon genes [6]. It is now well-established that post-transcriptional mechanisms involved in mRNA processing are deregulated in a multitude of diseases, including cancer. The impact and general role of alternate RNA splicing in tumorigenesis has only recently been uncovered, and though progress has been made on finding specific mechanisms and therapeutic strategies, more work is needed to more comprehensively understand the way malfunctioning splicing processes contribute to cancer phenotypes [7]. However, even in this early stage of exploration regarding alternative splicing (AS), its potential as a therapeutic target is starkly evident. Thus, a better understanding of these vulnerabilities and the identification of cancer-specific mRNAs, created by abnormal mRNA processing and modifications, would provide new strategies for cancer therapeutics.

Herein we aim to provide an overview of recent work and major themes regarding AS and its role as a disease mechanism and emerging therapeutic target in cancer, particularly in gastrointestinal (GI) malignancies. We then discuss the AS processing mechanisms that are being targeted by novel anticancer strategies, including small-molecule inhibitors and therapeutic oligonucleotides.

## 2. Basic Mechanisms of Alternative Splicing Regulation

Alternative splicing relies on the distinction between intronic and exonic sections of DNA within genes. The pre-processed mRNA transcript bears these same sections, which are recognized and spliced together by the spliceosome, a large complex of five small nuclear ribonucleoproteins (snRNPs) and proteins [8]. Specific consensus sequences such as 5′ dinucleotide GU and 3′ dinucleotide AG in introns are critical to intron recognition. In brief, actual splicing involves an enzyme-assisted lariat formation through attack of the 5′ splice site (SS) phosphodiester by the 2′ OH on a specific adenosine residue contained within the intron approximately 18–40 nucleotides upstream of the 3′ SS [9]. The freed 3′ OH of the 5′ SS then is able to attack the 3′ SS phosphodiesterase, leading to exon ligation and lariat release [9].

Another considerable layer of complexity arises when considering the propensity for a common gene to be spliced in different ways in different cells or even within the same cell, with varying exon inclusions and splicing [10]. While much remains to be learned about the regulatory mechanisms involved in this, a few have been uncovered.

The first of these are cis-acting elements along the pre-mRNA, which represent regulatory sequences facilitating everything from protein interaction with the pre-mRNA to folding and the three-dimensional structure of the molecule [11]. SSs themselves, in fact, fall under this category, setting the field initially to lay out the options upon which the spliceosome machinery can act. SS properties depend not only on the sites themselves, which remain highly conserved regions of the genome, but also on the surrounding sequences which have been found to increase or attenuate the binding interaction between recognition spliceosome snRNP U1 and the site [12]. This effect in fact allows the classification of SSs as “strong” or “weak”, with weak SSs typically flanking alternatively spliced exons (as opposed to constitutively spliced exons) [12].

Certain sequences on the pre-mRNA additionally can serve as binding sites for trans-acting RNA-binding proteins (RBPs), allowing for a higher level of granularity in promoting or inhibiting certain splicing events [13]. Modulated accessibility to RBPs or even the spliceosome itself through pre-mRNA folding has also been shown to have a significant regulatory effect, and conversely, RBPs may act directly by altering the structure of the pre-mRNA to promote or inhibit favorable spliceosome–SS interactions [14,15].

Two major classes of trans-acting RBPs are serine/arginine-rich proteins (SR proteins, often classed as “SRSF” for serine/arginine-rich splicing factor) and heterogeneous nuclear ribonucleoproteins (hnRNPs) [16,17]. SR proteins typically work by directly recruiting the spliceosome snRNP U1 to the 5′ SS or by recruiting U2AF, an auxiliary splicing factor, to the 3′ SS, leading to overall splicing enhancement [18]. In contrast, hnRNPs typically interact with intronic splicing silencer (ISS) motifs to avoid splicing at a specific SS [19]. However, many exceptions to this generalization have been uncovered, and both SR proteins and hnRNPs have been shown to both positively and negatively regulate splicing through binding various pre-mRNA motifs and cooperative and competitive direct interaction [19,20,21,22,23,24]. The phosphorylation of RBPs presents another means of modulating their activity and pre-mRNA binding effect [25]. Dysregulation of SR proteins or hnRNPs is a frequently observed trait in many GI malignancies.

Tissue-specific RBP expression also plays an integral role in the regulation of alternative splicing. Direct interactive effects between RBPs as well as the interplay of cis-element type and positioning along with the pre-mRNA transcript, chemical regulation (such as through phosphorylation), and physical and structural realities within the cellular environment serve to create a unique regulatory environment in different cell types for alternative splicing [26]. Such variance among different cell types allows alternative splicing to play a major contributory role in the determination of tissue identity and cell phenotype [26].

Because human alternative splicing typically occurs alongside transcription, certain properties of the gene transcriptional environment can also regulate alternative splicing. This is partly determined by indirect effects, such as the impact of transcription rate on the three-dimensional folding of the pre-mRNA transcript. However, this same rate has also been shown to have an impact on SS recognition, with slower rates leading to increased splicing at weaker splice sites, for instance, and faster rates favoring splicing at strong splice sites instead [11]. Such considerations have been termed the “kinetic model” of alternative splicing [27].

A “recruitment model”, which encompasses the direct recruitment of RBPs and other factors to the splicing environment, is also involved in the regulation of alternative splicing by transcriptional dynamics. Direct interaction between RNAP II and splicing factors, for instance, has been proposed as one model of modulating the splicing environment [28]. Moreover, this direct recruitment activity by RNAP II has been shown to affect transcription rates, as splicing machinery and various related factors are recruited to the pre-mRNA. As such, the relationship between transcription, splicing, timing, and present factors is incredibly dynamic, with many chances for cross-regulation and selectivity in determining the ultimate mRNA product to be translated [29].

Epigenetic factors on DNA have also been shown to influence regulating alternative splicing. While some of this is due to their influence on previously discussed methods—nucleosome positioning, for example, has an impact on the transcriptional rate and can cause RNAP II pausing [30]—interactions between splicing-related factors and epigenetic histone marks as well as nucleosomes themselves can contribute to the determination of which splicing factors are present for pre-mRNA processing and which are absent [31,32,33]. It follows that factors involved in these epigenetic marks, such as HDAC or even DNA modification, also play a role in alternative splicing [34]. DNA-binding proteins (DBPs) and influence over transcription (such as alternative reading frames) have also been seen to affect splicing, potentially through these same mechanisms, although work is still underway to more thoroughly explore this [28,35,36].

Finally, the spliceosome itself has been proposed to have a regulatory function on alternative splicing. Different points of control include spliceosome formation, the concentration of snRNP isoforms, and perhaps most interesting, kinetic proofreading [37]. (Indeed, snRNP differential expression among different tissues may be a clue as to the importance of this core regulatory function of the spliceosome [38].) Kinetic proofreading involves spliceosome rejection of an initially recognized SS, mediated by downstream catalytic steps within splicing having the ability to cancel the overall process based on chemical timing (most often timing inherent ATPase activity against catalytic activity) [39]. Overall, the ability for the spliceosome to self-regulate presents an interesting consideration within the larger discussion of alternative splicing and a tantalizing area for further work.

## 3. Splicing Abnormalities in Cancer

The perturbed regulation of alternative splicing events which result in the generation of multiple different isoforms and diversify protein expression is usually associated with tumorigenesis and is present in nearly all types of cancers (reviewed in [7,40,41,42]).

Tumor-specific splicing events confer a putative new class of alternative splicing-associated peptides as potential neoantigens which can affect the immune response and could be exploited as new targets in immunotherapy, such as in personalized tumor vaccines. For example, a recent systematic analysis of data from 8705 patients with one of thirty-two types of cancer revealed that tumors have up to 30% more alternative splicing events than normal tissues [43]. Moreover, tumors contain splicing isoforms that are not detected in tissues from healthy individuals, suggesting that novel, tumor-specific splicing events occur [43]. During the process of transformation of normal cells into certain cancer cells, alternative splicing might have a critical role [7]. The process usually includes escape from cell death and immune surveillance, cellular proliferation, de-differentiation, apoptosis avoidance, angiogenesis, invasion/metastasis, and energy metabolism through the regulation of the alternative expression of many oncogenic or tumor suppressor genes, in addition to splicing factors. Tumor cells can also acquire resistance to therapy after the generation of splicing variants [44,45].

As reviewed in a recent report [7], aberrant mRNA splicing has been shown to contribute to tumor progression as oncogenic drivers and/or bystander factors. Furthermore, the alterations in splicing factors found in tumors and other mis-splicing events such as long noncoding and circular RNAs have been shown to be contributing factors in tumorigenesis.

Recent therapeutic strategies targeting splicing catalysis and splicing regulatory proteins to modulate pathogenically spliced events as well as abnormally alternative splicing isoforms resulting in tumor-specific neoantigens for cancer immunotherapy are providing new opportunities for RNA-based therapies for the treatment of cancer.

## 4. Functions of Alternative Splicing in Gastrointestinal Malignancies

Given the vast network of regulatory interactions over alternative RNA splicing and the key role the process plays in tissue identity and gene expression, it is perhaps unsurprising that certain patterns have emerged when looking at alternative splicing within cancers. In fact, work on profiling alternative splicing in cancers, especially gastrointestinal malignancies, has seen a dramatic rise over recent years, uncovering a host of previously unknown disease mechanisms. A few of the major ones discovered in recent years are described below.

Xiong et al., in 2018, ran an analysis of RNA sequencing data from a cohort of patients with colorectal carcinoma (CRC) to identify differently expressed alternative splicing events (DEAS), finding a pattern of abnormal alternative splicing events in genes related to protein kinase activity, PI3K-Akt signaling, and p53 signaling, as well as a link between alternative splicing events and survival [46]. Mechanistic explanations for this phenomenon vary widely. SRSF6, an SR RBP, for example, is frequently upregulated in CRC and leads to aberrant splicing of ZO-1, causing oncogenic properties [47]. Indeed, database analysis reveals an association between SRSF6 overexpression and poor prognoses with higher rates of proliferation and metastasis [48]. SNHG6, an RNA gene, is also seen to have higher expression levels in CRC compared to normal tissue and has a positive correlation with poor prognosis [49]. Mechanistic workup suggests SNHG6 along with hnRNPA1 together lead to a favoring of PKM2 over PKM1 through alternative splicing, reprogramming CRC metabolism to enhance aerobic glycolysis among other proliferative effects [49]. Another hnRNP, hnRNPC, has also been shown to be associated with abnormal regulation favoring cell invasiveness and proliferative potential, and could be a player in CRC metastasis [50]. Stress, particularly nutrient starvation, can lead to PHF5A (a part of spliceosome snRNP U2) hyperacetylation, inducing alternative splicing that provides a more stable mRNA transcript of KDM3A and consequent overexpression, something which is attributed to stress resistance as well as colon carcinogenesis and invasiveness [51].

A different hnRNP, hnRNP K, has been implicated, along with SR RBPs SRSF1 and SRSF2 and hnRNP A1, in apoptotic dysregulation in pancreatic and liver cancers through the dysregulation of a host of target genes involved in both extrinsic and intrinsic apoptosis (including Fas, caspase-8, and caspase-9) as well as anti-apoptotic factors Bcl-x and Mcl-1. These RBPs are being currently explored for therapeutic targeting [52]. Overexpression of Linc01232 in pancreatic cancers (PCs) leads to the inhibition of hnRNP A2/B1 ubiquitination and degradation, leading to AS of A-Raf and thus dysregulation of MAPK/ERK signaling driving tumor progression [53]. Similar to CRC, pancreatic ductal adenocarcinoma (PDAC) has also been shown to favor the PKM2 isoform over PKM1 secondary to PTBP1 upregulation and increased incidence of PTBP1 pre-mRNA binding, particularly in drug-resistant PDAC (DR-PDAC) [54]. In the same study, knockdown of PTBP1 in vitro was shown to decrease PKM2 levels and sensitize cells to drug treatment [54]. PCs are also observed to be high in microRNA miR-193a-5p, linked to the disruption of AS through the targeting of splicing factors [55]. Specifically, miR-193a-5p overexpression is hypothesized to target SRSF6, leading to AS of OGDHL and ECM1, driving epithelial–mesenchymal transition (EMT) and increasing metastatic events [56]. Beyond specific molecular targets, a 2020 study of 177 patient PCs found the overall AS signature to have significant predictive prognostic power [57].

Among the discovery of similar disease mechanisms involving mutations of trans-factors such as SR proteins or upstream regulators of these factors, a study of alternative splicing in gastric cancers (GCs) recently led to the discovery of the importance of a class of circular noncoding RNAs (circRNAs). While these molecules have been known since as early as 1976, certain effects on alternative splicing promoting tumorigenesis have been described in recent literature. For one, the biogenesis of circRNAs is through the AS process, and therefore competes with normal AS [58]. Moreover, because circRNAs are noncoding, in addition to competing for splicing machinery, the biogenesis of a circRNA disables a potentially coding pre-mRNA, directly regulating gene expression [58]. Other effects, such as the regulatory activity of circRNAs on RNAP II leading to a change in transcriptional environment or association with snRNPs involved in the spliceosome, also affect the dynamics of AS, suggesting a possible disease mechanism explaining observed abnormal circRNA levels in GCs [59]. It should be emphasized, however, that general disease mechanisms related to circRNA are plentiful and its role with AS is only one of these pathways.

Disease mechanisms in other GI system cancers bear overall similarity to those described previously in CRC, PCs, and GCs, with individual efforts underway to profile the AS landscape and establish predictive links between AS events and prognosis. One such effort with hepatocellular carcinoma (HCC) in 2020, for instance, has identified over 3000 candidate AS events associated with almost 400 splicing factors, ultimately producing a predictive model for prognosis and metastatic potential [60]. The authors found, in particular, a strong correlation between YBX3 and prognosis as well as metastasis, proposing a mechanism through ABCA6 and PLIN5 and its effects on the primary bile acid biosynthesis pathway [60]. Esophageal squamous cell carcinoma (ESCC) analysis has revealed the role of long intergenic noncoding RNA (lincRNA) uc002yug.2 in carcinogenesis, particularly through the modulation of the nuclear AS environment to favor the RUNX1 isoform RUNX1a and reduce CEBPα, an event found to have predictive potential over prognoses in ESCC patients [61]. Interestingly, literature on gallbladder cancer (GBC)-related AS events is far sparser in comparison to other GI malignancies. However, circRNA, particularly circERBB2 overexpression, has been implicated in poor GBC prognoses and may provide a clue as to pathological AS events in such cancers in a similar manner as to GC, though as previously mentioned the broad scope of circRNA functions makes it difficult to narrow its impact to AS specifically [48].

Nevertheless, the role of AS on carcinogenesis and GI malignancies particularly is not to be understated. Promising preclinical work showing the therapeutic power of targeting aberrantly regulated players within this pathway suggests an emerging treatment strategy on the patient-facing front.

## 5. Cancer Therapeutics Targeting Aberrant RNA Splicing

Given the fact that cancer cells can display widespread changes in RNA splicing compared to normal cells, modulating RNA splicing in some cancer types might provide therapeutic benefits. Currently, potential therapeutic options mainly include immunotherapeutic avenues that exploit the immunogenicity of alternatively spliced protein products, small-molecule-mediated spliceosome modulation, splice-switching oligonucleotide (SSO)-based splicing regulation, and some RNA-based therapeutic approaches.

### 5.1. The Potential Role of Splicing for Cancer Immunotherapy

It is well-known that the antigenic presentation of endogenous cellular or exogenous viral protein-derived peptides on tumor cells by a major histocompatibility complex (MHC) can be recognized by T cells which may result in the rejection of tumor cells [62]. Important cancer immunotherapy approaches such as T-cell-receptor-engineered T cells (TCR-T cells) for adoptive cell therapy and therapeutic vaccines require the identification of appropriate target antigens. Thus, targetable tumor-specific antigens (TSAs) are crucial for enhancing the safety and efficacy of systemic immunotherapies [63]. Among different candidates, neoantigens derived from tumor-specific mRNA processing events including mRNA splicing, polyadenylation, and editing might have potential in this setting. In their large-scale analysis of 8656 tumor samples from The Cancer Genome Atlas (TCGA), Jayasinghe et al. [64] identified 1964 splice-site-creating mutations (SCMs). The same study suggests SCM-induced alternative splice forms are more immunogenic with a better T cell immune response and increased PD-L1 expression, supporting potential roles in cancer immunotherapy. Another comprehensive analysis of alternative splicing across 32 TCGA cancer types from 8705 patients by reanalyzing RNA and whole-exome sequencing data detected tumors with up to 30% more alternative splicing events than in normal samples [43]. The same study suggested that predicted neoepitopes formed by tumor-specific mRNA splicing events are more frequent than those formed by somatic single-nucleotide variants (SNVs). Although the recent data are promising for the potential role of mRNA splicing in cancer immunotherapy (Figure 1), functional studies to validate tumor immunogenicity and test the possible benefits of therapeutic interventions are warranted.

### 5.2. Small-Molecule Modulators of the Spliceosome in Cancer

Several natural products and their synthetic derivatives display antitumor activities by binding to components of the spliceosome involved in the removal of introns from mRNA precursors in eukaryotic cells (Figure 2). The screening of natural products derived from bacteria, called pladienolides, herboxidienes, and FR901464, has resulted in potent compounds with antitumor activity [65,66,67,68,69,70,71,72,73,74,75] and led to the development of their synthetic analogs with improved stability, solubility, and activity. These include the pladienolide derivative E7107, the FR901464 derivatives spliceostatin A and meayamycins, and sudemycins, which possess a pharmacophore that is common to FR901464 and pladienolide. Although structurally different, these drugs were shown to modulate alternative RNA splicing by targeting the SF3b1 subunit (a five-polypeptide subcomplex of the U2 snRNP) spliceosome, which led to further research to modulate the spliceosome for cancer therapy.

As a result of these developments, small molecules that target splicing factors have been developed for various cancers including GI malignancies [67,68]. Spliceostatin A (SSA) is one of the early splicing modulators as a natural compound with known antitumor effects on murine colon tumors (colon 38) and ability to inhibit splicing by combining with SF3b [69,70]. With similar anti-splicing effects, other splicing modulators such as meayamycin B (MAMB) and pladienolides were later shown to inhibit tumor growth of human colorectal carcinoma and gastric cancer cell lines [71,72,73,74]. The pladienolide analog E7107 blocks spliceosome assembly by preventing tight binding of U2 snRNP to pre-mRNA and was tested in phase 1 studies with patients presenting different types of solid tumors, including colorectal, esophageal, gastric, and pancreatic, and was found to stabilize tumor growth [75,76,77]. However, the development of E7107 was suspended after the incidence of two cases of vision loss during the trial, likely related to E7107. Different than E7107, H3B-8800, an orally available modulator of the SF3b complex, was found to be highly selective for cells harboring a mutant Sf3b1 gene, not only in hematopoietic cells but also in solid tumor cell lines, including the pancreas and colon [78]. The results of the phase 1 study using H3B-8800 in myeloid neoplasms have recently been published [79], but there is no existing clinical trial using this molecule in GI malignancy. By using human hepatocellular carcinoma cell line Huh-7 in preclinical studies, it was shown that small-molecule amiloride could “normalize” the splicing of BCL-X, HIPK3, and RON/MISTR1 transcripts [80].

#### Clinical Trials of Small-Molecule Splicing Inhibitors

The use of RNA splicing modulators in clinical trials appears as an attractive treatment option for establishing novel therapeutic cancer drugs. Towards this goal, several clinical trials, mostly targeting PRMT for various cancers including GI malignancies, are currently underway, but these are still in their early phases (Table 1). Protein arginine *N*-methyltransferase 5 (PRMT5) is the predominant type II PRMT which is critical in the assembly of crucial components of the spliceosome, snRNPs [81]. There are several early clinical trials that are testing different PRMT inhibitors for a variety of cancers [82]. The first in-human phase 1 trial with JNJ-64619178, an inhibitor of PRMT5, in patients with advanced patients that have solid tumors is active but not recruiting as of September 2021 (NCT03573310). GSK3326595 is another PRMT5 inhibitor in a phase 1 study (Meteor 1) (NCT02783300) including patients with advanced or recurrent solid tumors. This is a three-part study where part one is a dose escalation, part two is the disease-specific expansion, and part three is the dose determination of GSK3326595 in combination with pembrolizumab. Part one results were presented as showing a manageable safety profile with signs of some activity in multiple tumor types, including colorectal cancer (17%) [83]. The disease-specific expansion includes a variety of solid tumors but not GI tumors (HPV+ solid tumors of any histology included). A recent preclinical study using the same inhibitor showed that GSK3326595 inhibits the growth of liver tumors in human-MYC-overexpressing transgenic mice that spontaneously develop HCC [84]. Moreover, the combination of GSK3326595 with anti-PD1 therapy improved the efficacy, which might be worth testing in future HCC clinical trials. Another preclinical study tested a variety of human cancer cell lines and showed the anti-proliferative activity of GSK3326595. Interestingly, the study showed that the inhibition of PRMT5 activates the p53 pathway via the induction of alternative splicing of MDM4 [85]. PF-06939999 is a selective small-molecule inhibitor of PRMT5, which is in a phase 1 study which is actively recruiting patients with advanced/metastatic solid tumor types marked by potential frequent splicing factor mutations, including esophageal cancer (NCT03854227). The preliminary activity of the inhibitor presented recently shows an acceptable safety profile with objective tumor responses in patients with head and neck squamous cell carcinoma (HNSCC) and non-small-cell lung cancer (NSCLC) [86]. Enrollment to part two, dose expansion, of the study is ongoing. Among several PRMT5 inhibitors, PRT811 was shown to be brain penetrant in preclinical studies [87], and an open-label phase 1 study in patients with advanced cancers including solid tumors, CNS lymphoma, and/or high-grade gliomas is actively recruiting (NCT04089449). Another potent PRMT5 inhibitor with preclinical activity, PRT543 [88], is actively being tested in a phase 1 study in patients with advanced solid and hematologic malignancies (NCT03886831). A preclinical study testing a potent, type I PRMT inhibitor, GSK3368715 (EPZ019997), showed cytotoxicity mostly in hematologic malignancies and in a subset of solid tumor cell lines, including 13% of pancreatic cancer [89]. Mtap gene deficiency was shown to impair PRMT5 activity, sensitizing cancer cells to GSK3368715, and inhibition of PRMT5 produced synergistic cancer cell growth inhibition when combined with GSK3368715 in the same study. Using GSK3368715 in the first-in-human phase 1 study of patients with solid tumors and DLBCL completed the recruitment in March 2021 (NCT03666988). More clinical data with late-phase clinical trials are needed to provide more efficacy and safety data.

By using sulfonamides such as indisulam (also known as E7070), tasisulam, and E7820 (aryl-sulfonamides) in both preclinical and clinical studies, activity against solid tumors, including GI malignancies, were previously shown in multiple studies [90,91,92,93,94,95,96,97,98]. They promote the degradation of the splicing factor RBM39, which induces intron retention and exon skipping, but their cellular mechanism of action was not fully understood for many years despite their known anticancer properties [99,100,101]. Thus, it might be worth exploring the clinical utility of these compounds in appropriate cancer patient populations in future clinical trials.

### 5.3. Splice-Switching Oligonucleotides

Oligonucleotide-based therapies can directly modulate pre-mRNA splicing through allowing selective induction and regulation of splice site specificity (Figure 3). Splice-switching oligonucleotides (SSOs) are 15–30-nucleotide-long synthetic oligonucleotide molecules comprised of nucleotides or nucleotide analogs designed to bind to a complementary pre-mRNA sequence through Watson–Crick base pairing and create a steric block to the binding of splicing factors to the pre-mRNA, which alters the recognition of splice sites by the spliceosome, leading to a modification of normal splicing of the targeted transcript [102]. Thus, this technology can be used as a therapeutic intervention that can induce degradation or interfere with the splicing of pre-mRNA.

Recently, the Food and Drug Administration (FDA) has approved the SSOs eteplirsen for the treatment of Duchenne muscular dystrophy [103] and nusinersen for the treatment of spinal muscular atrophy [104], respectively. Although SSOs in cancer therapy are still under development, some promising preclinical results are emerging (Table 2). As an example, a recent study showed that SSOs can promote MDM2-ALT1 splicing and induce p53 protein expression, as well as apoptosis in p53 wild-type cells [105].

Alternative splicing of PKM contributes to the control of glucose metabolism by producing either the PKM1 isoform which contains exon 9 and leads to oxidative phosphorylation or the PKM2 isoform which contains exon 10 and leads to aerobic glycolysis; that is, the Warburg effect [106]. It has been reported that the isoform PKM2 is commonly expressed in various cancers, and SSOs that interfered with the expression of PKM2 promoted apoptosis in glioblastoma cell lines [107].

SSOs have also been applied to produce autoinhibitory HER2 protein isoforms by the modification of HER2 pre-mRNA alternative splicing in breast cancer cells [108]. Likewise, SSOs have been used to correct splicing defects caused by a deletion polymorphism in intron 2 of the *BIM* gene that is associated with resistance to imatinib cancer therapy in chronic myeloid leukemia [109]. Accumulating literature supports the use of SSOs for several other targets, which are highlighted in Table 2.

**Table 2 ijms-22-11790-t002:** Preclinical studies using splice-switching oligonucleotides in cancer.

Target (Pre-mRNA)	In Vitro and In Vivo Model Systems	Functional Splicing Outcome	References
BCLX (i.e., BCL2L1)	Breast, cervical, prostate, and glioma cell lines, and melanoma tumor xenografts	Isoform switch from anti-apoptotic BCL-XL to pro-apoptotic BCL-XS protein.	[110,111,112]
BIM (i.e., BCL2L11)	CML cell lines	Blocking exon 3 but enhancing exon 4 splicing, thereby resensitization of BIM deletion-containing cancer cells to imatinib.	[109]
BRCA1	Breast cancer cell line	Artificially stimulating skipping of exon 11 in endogenous BRCA1 pre-mRNA, promoting DNA double-strand breaks and therefore causing synthetic lethality (more susceptibility to PARP inhibitors).	[113]
ERBB2 (i.e., HER2)	Breast cancer cell lines	Inducing skipping of exon 15 in HER2 pre-mRNA, leading to the upregulation of Δ15HER2 mRNA, which has autoinhibitory activity.	[108]
ERBB4 (i.e., HER4)	Breast cancer cell lines and tumor xenografts	Directing the alternative splicing of HER4 from the CYT1 to the CYT2 isoform with an inhibitory effect on cancer cell growth.	[114]
IN-RA	Rhabdomyosarcoma-derived cell lines	Impeding the IGF2 pathway by reducing IN-RA expression (targeting the exon-11-skipped IN-RA isoform) and consequently mitigating cancer cell proliferation, migration, and angiogenesis.	[115]
MDM2	Rhabdomyosarcoma and breast cancer cell lines	Blocking the exon 11 SRSF2 binding sites promoted MDM2-ALT1 splicing and induced p53 protein expression and apoptosis in p53 wild-type cells.	[105]
MDM4	Melanoma cell lines, melanoma, and DLBCL PDX mice models	Induced skipping of exon 6 leading to decreased MDM4 abundance, thereby inhibiting tumor growth and enhancing sensitivity to MAPK-targeting therapeutics.	[116]
MKNK2	Glioblastoma, hepatoma, and breast cancer cell lines	Isoform switch from oncogenic Mnk2b protein to tumor-suppressive Mnk2a isoform.	[117]
PKM	Glioblastoma cell lines	Targeting the enhancer in exon 10 and switching the splicing of endogenous PK-M transcripts to include exon 9, thereby leading to apoptosis.	[107]
STAT3	Melanoma, breast, lung, prostate cell lines, and breast cancer xenografts	Targeting a splicing enhancer that regulates STAT3 exon 23 alternative splicing specifically, promoting a shift of expression from STAT3α to STAT3β, leading to apoptosis and cell cycle arrest.	[118]

### 5.4. Other Therapeutic Approaches

Modulating some other splicing targets, such as SmgGDS, MKNK2 in cancer cells was shown to inhibit tumor development in preclinical studies [55,117]. Given the fact that cancer cells are characterized by high telomerase activity compared to normal cells which express little or no telomerase, telomerase is another potential target for cancer treatment. Modulating the alternative splicing of human telomerase reverse transcriptase (hTERT), pre-mRNA was shown to inhibit telomerase activity and thereby decrease cell proliferation and induce apoptosis in glioma cells [119], which suggests another potential target in future studies. The M2 pyruvate kinase (PKM2) isoform is an example of a target for the metabolic state of cells. A recent preprint study showed that surrogate mouse-specific ASO induces PKM splice switching and inhibits tumor growth in a genetic HCC mouse model without toxicity [120]. An alternative technology using sense oligonucleotides that bind to RNA-binding proteins rather than RNA was also tested [121]. Using decoy oligonucleotides that target splicing factors RBFOX1/2, SRSF1 and PTBP1, the study showed that decoy oligonucleotides can specifically bind to their respective splicing factors and inhibit their splicing and biological activities both in vitro and in vivo. Another promising technology is clustered regularly interspaced short palindromic repeats (CRISPR)-Cas and related systems, as splicing manipulation techniques by genome or RNA editing [122]. This technology has potential use in cancer therapy in future clinical trials. Other technologies such as short hairpin RNA interference, small interference RNA, single-base editors (BEs), or cytosine-based editors (CBEs) are also among exciting strategies which will likely be tested in future clinical studies [123,124,125,126,127].

## 6. Conclusions and Future Perspectives

Recent large-scale genome, transcriptome, proteome, and epigenome profiling efforts and functional characterization of the candidate factors involved in RNA splicing have identified numerous abnormal RNA splicing events that have been implicated in tumorigenesis as oncogenic drivers and/or passengers in various cancers. Alternative splicing can produce multiple isoforms and pathologic variants with diverse functions for the same gene loci, and dysregulation of alternative splicing of mRNA has been implicated in the pathogenesis of different types of cancers. Because of this, the area of alternative splicing has become an attractive topic for cancer research to elucidate the regulatory mechanisms of alternative splicing and to further our understanding of tumorigenesis, which is also leading to new therapeutic strategies. As a result, therapies including small molecules that modulate splicing, SSOs, as well as novel RNA-based CRISPR-Cas13a editing technology and others that target abnormal splicing sites or events have also been explored as novel therapeutic strategies for various diseases, including cancer.

The identification of neoantigens that elicit a specific immunogenic response is key for the development of effective cancer vaccines. However, many challenges remain before many of the cancer-specific candidate TAs can be translated into effective therapies. In addition, there remain a few challenges for targeting the abnormal splicing events and/or aberrant RNA species, which are the specificity and delivery efficiency which must be addressed before these strategies become clinically meaningful. Nevertheless, neoantigens derived from somatic mutations and alternative RNA splicing have been extensively characterized as a source of Tas, and they represent a novel immune therapeutic strategy that is still under investigation.

The development of small-molecule drugs that target highly structured elements in aberrantly processed disease-causing mRNAs has been explored [128,129]. These efforts provide new opportunities for identifying new, druggable binding sites in pre-mRNAs involved in many cancer types. It has been demonstrated that small molecules can bind specific structural conformations within introns to induce structural changes that modulate alternative RNA splicing and gene expression [130].

Although accumulating literature indicates that aberrant regulation of splicing is involved in tumorigenesis, the role of splicing in cancer pathogenesis, particularly in GI cancers, has not been fully elucidated. The targeting of splicing may provide novel attractive methods of treating cancer; however, the specificity and delivery efficiency are among the major challenges facing scientists. It should also be noted that mutations in genes related to splicing are rare in solid tumors, in contrast to in hematologic malignancies. Ongoing clinical studies are of great importance as they may provide new insights into splicing dysregulation in solid tumors and improve RNA-based anti-tumor therapy in the near future. Furthermore, although it is not the focus of this review article, the dysregulation of pre-mRNA processing, including alternative RNA splicing, provides a potentially extensive source of therapeutic antigens (TAs) for targeted immunotherapy. Post-transcriptional dysregulation contributes to the antigen profile of tumors, and this contribution has been leveraged for immunotherapy and led to the development of mRNA cancer vaccines which are attracting attention as SARS-CoV-2 mRNA-based vaccines have shown their feasibility, effectiveness, and scalability. Emerging preclinical and clinical data demonstrate that mRNA cancer vaccines are safe and efficient with the potential for rapid, inexpensive, and scalable manufacturing [131,132,133,134].

Despite the relative infancy of the field and inherent challenges involving specificity and delivery, therapeutic strategies targeting cancer-specific AS abnormalities are a promising, novel strategy for preclinical and clinical research, with potentially impactful clinical outcomes. As current research in the understanding of splicing abnormalities, addressing challenges involving specificity and delivery, and ongoing clinical trials progress, increasingly more specific and effective therapies with impactful clinical outcomes are likely to emerge.

## Figures and Tables

**Figure 1 ijms-22-11790-f001:**
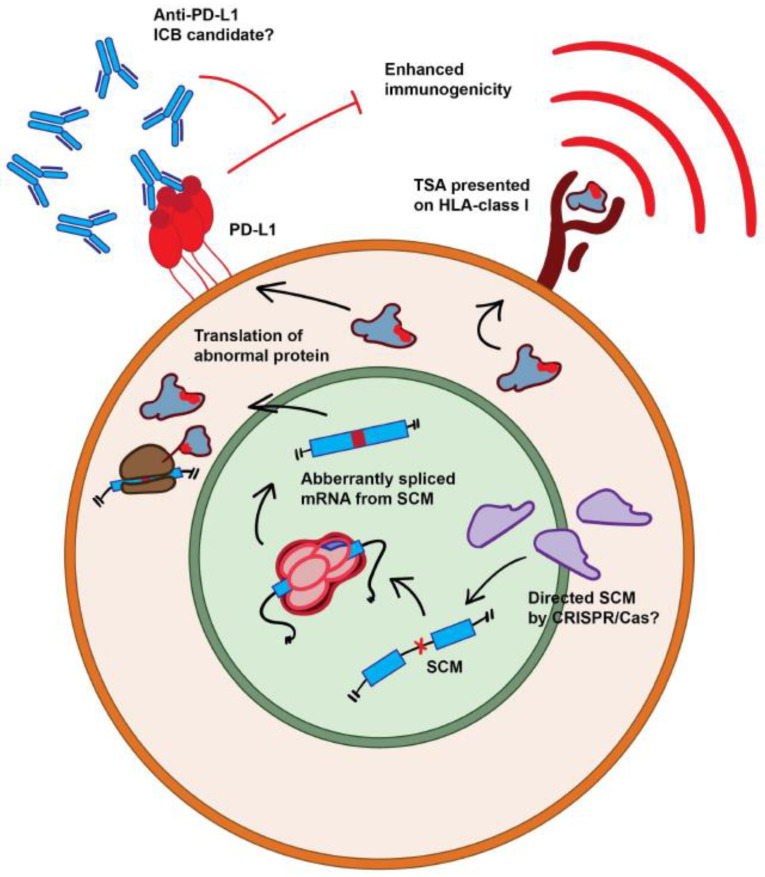
**Immunogenic effects of alternative splicing and immune-based therapy options targeting aberrant alternative splicing.** Mutations creating novel splice sites (SCM), either endogenic or induced by targeted CRISPR/Cas-based gene editing results in the creation of tumor-specific antigens (TSAs) through the translation of abnormally spliced RNA. Processing and presentation on HLA class I leads to enhanced T cell immunogenicity. The same process has also been implicated in the upregulation of PD-L1, potentially making such tumors candidates for immune checkpoint blockading (ICB) as PD-L1 typically suppresses immune activation. Red blunt arrows show negative regulation.

**Figure 2 ijms-22-11790-f002:**
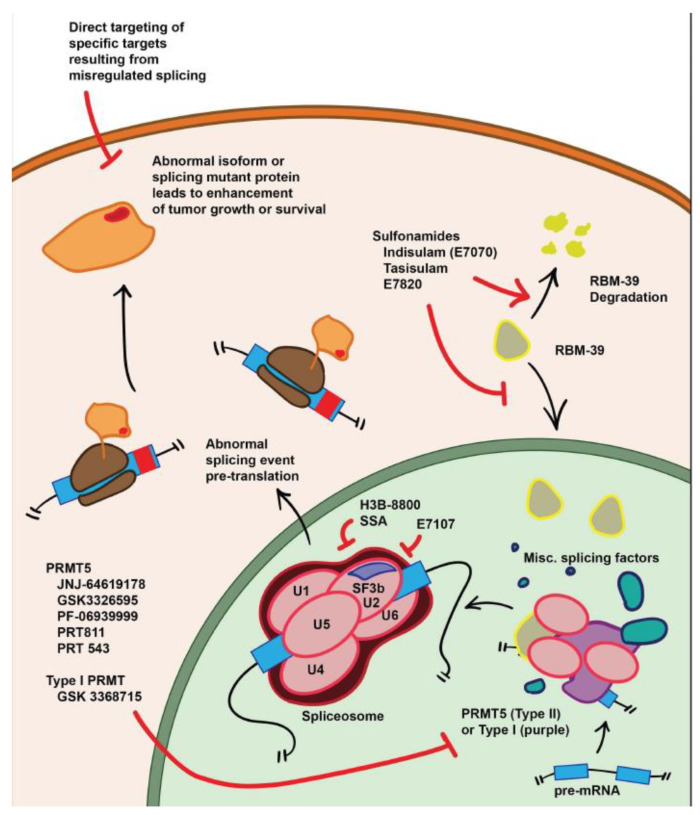
**Small-molecule-based therapeutic strategies targeting the alternative splicing environment.** Various components of the cellular machinery mediating alternative RNA splicing can and are being targeted by small molecules to restore functionality. Common targets include PRMT5 or various type I PRMTs, snRNP U2, SF3b, and RBM-39. Abnormally expressed isoforms or aberrant proteins exhibiting gain-of-function effects enhancing tumor proliferation or survival can also themselves be the subject of small-molecule targeting. Abbreviations: pre-mRNA, pre-messenger RNA; PRMT, protein arginine methyltransferase; RBM-39, RNA-binding protein 39; SF3b, spliceosome factor 3b; and snRNP, small nuclear ribonucleoprotein. Red blunt arrows show negative regulation/inhibition.

**Figure 3 ijms-22-11790-f003:**
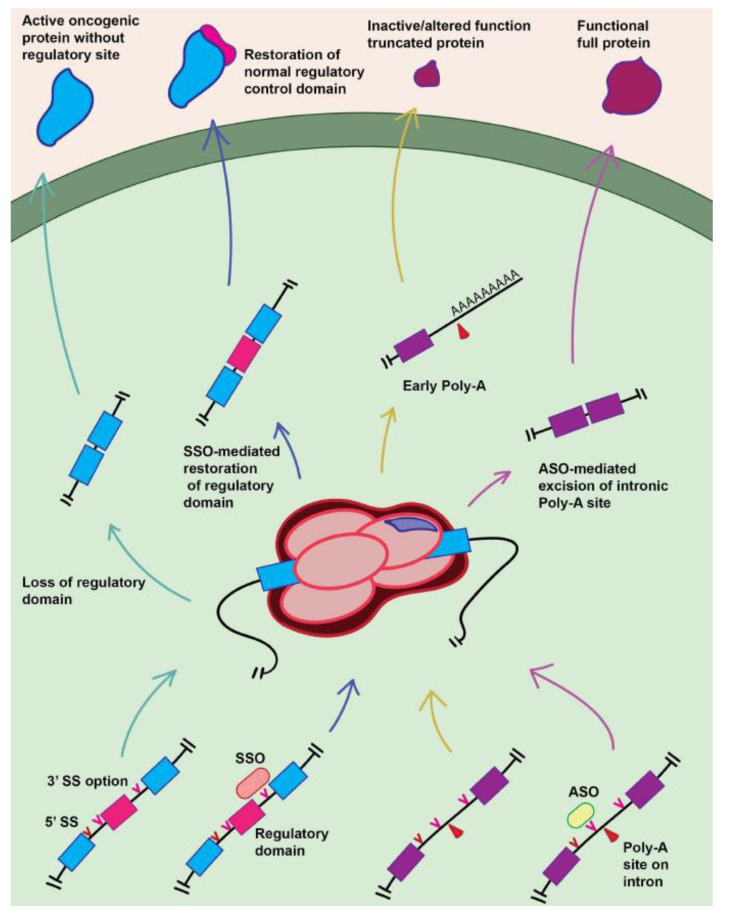
**Oligonucleotide-based therapeutic modulation of splice site selectivity by endogenous splicing machinery.** The presence of multiple different splice site options that all produce a viable mRNA transcript post-processing opens the possibility of incorrect selection of the correct splice site for the tissue or cell, with potential tumorigenic effects. SSOs can be used to bind splice sites and prevent recognition, therefore allowing the modulation of the produced protein. For instance, the figure shows a case where dysregulated splicing pathways cause incorrect splice site recognition, leading to the production of an oncogenic protein lacking a regulatory site (and thus presumably being constitutively active, promoting uncontrolled growth). The introduction of SSOs restores the proper splicing product, producing a normally controlled protein. More generally, ASOs can be used for a similar function. The figure depicts a case where dysfunctional splicing machinery leads to improper inclusion of an intron which contains within it a polyadenylation site, leading to premature polyadenylation and truncation of the product. Introduction of the ASO blocks recognition of this splice site that was allowing for the inclusion of the intronic polyadenylation site, restoring the normal protein product. Abbreviations: ASO, antisense oligonucleotide; poly-A, polyadenylation; SS, splice site; SSO, splice-switching antisense oligonucleotide.

**Table 1 ijms-22-11790-t001:** Small-molecule modulators of the spliceosome in ongoing cancer clinical trials (access date: October 2021).

Trial Identifier (ClinicalTrials.gov)	Phase	Status	Patient Characteristics	Drug and Treatment Regimen	Target
NCT03573310	1	Active, not recruiting	Advanced solid tumors, NHL, or lower risk MDS	JNJ-64619178 (po) monotherapy	PRMT5
NCT02783300	1	Recruiting	Advanced solid tumors and non-Hodgkin lymphoma	GSK3326595 (po) monotherapy; part 3 includes in combination with pembrolizumab	PRMT5
NCT03854227	1	Recruiting	Advanced or metastatic solid tumors	PF-06939999 (po) alone or in combination with docetaxel	PRMT5
NCT04089449	1	Recruiting	Advanced solid tumors and high-grade gliomas	PRT811 (po) monotherapy	PRMT5
NCT03886831	1	Recruiting	Advanced solid tumors and hematologic malignancies	PRT543 monotherapy	PRMT5
NCT03666988	1	Completed	Advanced solid tumors and diffuse large B cell lymphoma	GSK3368715 monotherapy	PRMT1
NCT028`41540	1/2	Recruiting	Myelodysplastic syndromes, acute myeloid leukemia, and chronic myelomonocytic leukemia	H3B-8800 monotherapy	SF3B
NCT03614728	1/2	Recruiting	Myelodysplastic syndromes and acute myeloid leukemia	GSK3326595 monotherapy	PRMT5

## Data Availability

Not applicable.

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
