# Peer review of "Therapeutic Targeting of Alternative RNA Splicing in Gastrointestinal Malignancies and Other Cancers"

_ijms, 2021, doi:10.3390/ijms222111790_

Round 1

Reviewer 1 Report

IJMS-1429312

Drs. Sahin et al have written a timely and comprehensive review of current efforts to target alternative mRNA splicing (AS) for therapeutic gain, focusing primarily on gastrointestinal (GI) malignancies.  The review is generally well-written, with appropriate section headings, figures and tables, but some of the figures and tables need to be better placed and discussed in the text; modifications are necessary for clarity and better flow of the authors’ ideas.  Specifically,

  1. Under section 5 (cancer therapeutics targeting aberrant RNA splicing), lines 289-291 describe all the potential therapeutic options to be discussed subsequently (neoantigens, small molecules, SSOs, RNA-based therapeutics), and the authors reference Fig 1. But Fig 1 only deals with subsection 5.1, the potential role of splicing for cancer immunotherapy.  Accordingly, this sentence (289-291) needs clarification, and Fig 1 should be referenced in subsection 5.1 and moved into this subsection.
  2. Similarly, in subsection 5.2, small molecule modulators of the spliceosome, this section should begin with a reference to Fig 2, and Fig 2 should be moved here; it’s out of place after Table 1.
  3. Additionally, the references in line 326 should be expanded from [63, 64] to [63-72], as the subsequent sentences and the next paragraph (lines 327-352) deal with the observations in those references.
  4. Discussion of the small molecule clinical trials starts on line 353 without warning. The authors should insert a sub-sub-section (5.2.a) “clinical trials of small molecule splicing inhibitors” in front of line 353, and make reference to Table 1 in an introductory sentence to this section.  Table 1 should be moved here, followed by the text.
  5. Lines 394-401: why did the authors not include these sulfonamide clinical trials in Table 1? Seems that they should be in the Table.
  6. Subsection 5.3 (splice-switching oligonucleotides [SSOs]) should begin (line 413) with an introductory sentence and reference to Fig 3. The introductory sentence could be the same as the title of Fig 3.  And Fig 3 should be moved to the front of this section.  The nice summary Table 2 could then follow the text as it does now.
  7. Section 6 (conclusions and future perspectives). It seems to this reviewer that the last paragraph (lines 514-522) should be moved between lines 490 and 491, as this fits with the flow of the overall text. 
  8. The overall conclusions section will need a summary sentence or two. As presented, and even after moving the last paragraph (as discussed in #7, above) the section ends abruptly. 
  9. Last, while not discussed, it would be useful if the authors give some thought and text, perhaps in the beginning of section 3 (possibly after line 193), to why tumors have perturbed and increased splicing events, splicing isoforms, and generally upregulated activities and amounts of splicing factors.

Overall, the authors have provided a timely and comprehensive review of current therapeutic approaches to interfere with aberrant splicing in tumors, with an emphasis on GI tumors; this should be of use to our community.  While the above comments are mostly editorial, and not conceptual, they should be considered in order to make this a very strong and useful review.

Author Response

We would like to thank you for all your comments and suggestions on our manuscript entitled " Therapeutic Targeting of Alternative RNA Splicing in Gastrointestinal Malignancies and other cancers."

The comments and feedback were very constructive and helpful in revising the manuscript to convey more clearly our work. We provide below point-by-point detailed responses to each comment and the corresponding changes to improve the manuscript. The manuscript in its current form is much more improved.

Please see the attachment for a point-by-point response to the reviewer’s comments.

Reviewer 1

Drs. Sahin et al have written a timely and comprehensive review of current efforts to target alternative mRNA splicing (AS) for therapeutic gain, focusing primarily on gastrointestinal (GI) malignancies.  The review is generally well-written, with appropriate section headings, figures, and tables, but some of the figures and tables need to be better placed and discussed in the text; modifications are necessary for clarity and better flow of the authors’ ideas.  Specifically,

1- Under section 5 (cancer therapeutics targeting aberrant RNA splicing), lines 289-291 describe all the potential therapeutic options to be discussed subsequently (neoantigens, small molecules, SSOs, RNA-based therapeutics), and the authors reference Fig 1. But Fig 1 only deals with subsection 5.1, the potential role of splicing for cancer immunotherapy.  Accordingly, this sentence (289-291) needs clarification, and Fig 1 should be referenced in subsection 5.1 and moved into this subsection.

1-A: We thank reviewer for the comment. The reference for Figure 1 has accordingly been moved to line 331 highlighting the potential role of mRNA splicing in cancer immunotherapy (line numbers reflect revised Word document with simple markup of tracked changes). Sentence 295-298 has also been reworded to better serve in establishing the major therapeutic avenues we explore in section 5 and through all three figures.

2- Similarly, in subsection 5.2, small molecule modulators of the spliceosome, this section should begin with a reference to Fig 2, and Fig 2 should be moved here; it’s out of place after Table 1.

2-A: The reference to Figure 2 has now been moved to line 349, accompanying the first sentence of subsection 5.2 which discusses the small molecules depicted in figure 2. The figure itself has accordingly been moved from after Table 1 to directly before 5.2, in close proximity to the reference and matching the style of subsection 5.1.

3- Additionally, the references in line 326 should be expanded from [63, 64] to [63-72], as the subsequent sentences and the next paragraph (lines 327-352) deal with the observations in those references.

3-A: The reference in line 351 (due to other edits 326 is now 352) has been accordingly updated to encompass all relevant sources used in subsequent sentences and paragraphs. Due to other edits, the reference numbers are now #65-75.

4- Discussion of the small molecule clinical trials starts on line 353 without warning. The authors should insert a sub-sub-section (5.2.a) “clinical trials of small-molecule splicing inhibitors” in front of line 353 and make reference to Table 1 in an introductory sentence to this section.  Table 1 should be moved here, followed by the text.

4-A: We added a sub-sub section”5.2.a. Clinical trials of small-molecule splicing inhibitors” with an introductory sentence to this section and made the reference to Table 1. This section’s header is in line 380.

5- Lines 394-401: why did the authors not include these sulfonamide clinical trials in Table 1? Seems that they should be in the Table.

5-A: There seems to be some confusion in Table 1 and we thank the reviewer for bringing this to our attention. Sulfonamide trials were performed many years ago and Table 1 aimed at summarizing ongoing clinical trials. Thus, the title was modified accordingly (added the word ‘ongoing’ and wrote access date; lines 443-444).

6- Subsection 5.3 (splice-switching oligonucleotides [SSOs]) should begin (line 413) with an introductory sentence and reference to Fig 3. The introductory sentence could be the same as the title of Fig 3.  And Fig 3 should be moved to the front of this section.  The nice summary Table 2 could then follow the text as it does now.

6-A: A new sentence to introduce section 5.3 has now been added (lines 489-490), with a reference to Figure 3. Figure 3 has accordingly been moved to accompany this reference and is now located directly prior to section 5.3. As suggested, no changes have been made to the placement of Table 2.

7- Section 6 (conclusions and future perspectives). It seems to this reviewer that the last paragraph (lines 514-522) should be moved between lines 490 and 491, as this fits with the flow of the overall text.

7-A: This paragraph has been moved as suggested and is now between lines 632-640.

8- The overall conclusions section will need a summary sentence or two. As presented, and even after moving the last paragraph (as discussed in #7, above) the section ends abruptly.

8-A: A couple of sentences summarizing the overall review as well and consequences and importance of the topic have been added (lines 664-669) to smooth the end of the section and the overall review.

9- Last, while not discussed, it would be useful if the authors give some thought and text, perhaps at the beginning of section 3 (possibly after line 193), to why tumors have perturbed and increased splicing events, splicing isoforms, and generally upregulated activities and amounts of splicing factors.

9-A: We added a few sentences with new references to the paragraph (Lines 196-200) to address points raised by the reviewer: “During the process of transformation of normal cells into certain cancer cells, alternative splicing might have a critical role [7]. The process usually includes escape from cell death and immune surveillance, cellular proliferation, differentiation, apoptosis, angiogenesis, invasion/metastasis, and energy metabolism through the regulation of the alternative expression of many oncogenic or tumor suppressor genes, and splicing factors. Tumor cells can also acquire resistance to therapy after the generation of splicing variants [44,45].

Overall, the authors have provided a timely and comprehensive review of current therapeutic approaches to interfere with aberrant splicing in tumors, with an emphasis on GI tumors; this should be of use to our community.  While the above comments are mostly editorial, and not conceptual, they should be considered in order to make this a very strong and useful review.

We thank the reviewer for the very constructive feedback. The manuscript in its current form is much more imroved.

Reviewer 2 Report

  Sahin et al. present a thorough and informative review centered on the role of alternative splicing alterations in gastrointestinal cancers.  The authors do a good job of discussing the mechanism of alternative splicing and the potential results this has in disease.  This is then followed by a discussion of how alterations in alternative splicing are related to GI cancers.  The authors close with a complete discussion of current therapeutic options in development or in clinical trial designed to exploit alternative splicing alterations in cancer as well as other diseases.  While a detailed discussion of certain mechanisms influencing alternative splicing is missing and could provoke further debate and advancement in the field, the authors have done a very good job of reviewing what is currently known.  This review should be of elevated interest to both those studying GI cancers as well as those studying alternative splicing.

This is a very thorough and interesting review.  My complements to the authors.

Author Response

POINT BY POINT RESPONSE TO REVIEWERS:

We would like to thank you for all your comments and suggestions on our manuscript entitled " Therapeutic Targeting of Alternative RNA Splicing in Gastrointestinal Malignancies and other cancers."

The comments and feedback were very constructive and helpful in revising the manuscript to convey more clearly our work. We provide below point-by-point detailed responses to each comment and the corresponding changes to improve the manuscript. The manuscript in its current form is much more improved.

Reviewer 2

Sahin et al. present a thorough and informative review centered on the role of alternative splicing alterations in gastrointestinal cancers.  The authors do a good job of discussing the mechanism of alternative splicing and the potential results this has in disease.  This is then followed by a discussion of how alterations in alternative splicing are related to GI cancers.  The authors close with a complete discussion of current therapeutic options in development or in clinical trial designed to exploit alternative splicing alterations in cancer as well as other diseases.  While a detailed discussion of certain mechanisms influencing alternative splicing is missing and could provoke further debate and advancement in the field, the authors have done a very good job of reviewing what is currently known.  This review should be of elevated interest to both those studying GI cancers as well as those studying alternative splicing.

This is a very thorough and interesting review.  My compliments to the authors.

A: We thank reviewer 2 for the very constructive feedback.
